# A Comparative Analysis between Subpectoral versus Prepectoral Single Stage Direct-to-Implant Breast Reconstruction

**DOI:** 10.3390/medicina56100537

**Published:** 2020-10-13

**Authors:** Jeong-Hoon Kim, Seung Eun Hong

**Affiliations:** Department of Plastic and Reconstructive Surgery, Ewha Womans University Mokdong Hospital, College of Medicine, Ewha Womans University, 1071 Anyangcheon-ro, Yangcheon-gu, Seoul 07985, Korea; kimsbrothers@hanmail.net

**Keywords:** acellular dermal matrix, breast reconstruction, subpectoral, prepectoral

## Abstract

*Background and objectives:* Until now subpectoral breast reconstruction (SBR) has been the predominant form; however, it can present with pectoralis muscle contraction and animation deformity. To avoid these complications, surgeons have begun placing breast implants in the same anatomic space as the breast tissue that was removed. We report a comparative analysis of prepectoral breast reconstruction (PBR) versus subpectoral breast reconstruction to analyze their differences. *Materials and Methods:* Direct-to-implant (DTI) reconstruction using acellular dermal matrix (ADM) performed from February 2015 to February 2020 were retrospectively reviewed. We then compared the clinical course and postoperative outcomes of the two groups (prepectoral vs. subpectoral) based on the overall incidence of complications, pain scale, and the duration of drainage. *Results:* A total of 167 patients underwent unilateral DTI, with SBR 114 (68.3%) and PBR 53 (31.7%). Patient demographics were similar between the two groups. There was no statistically significant difference in rates of seroma, infection (requiring intravenous antibiotics), hematoma, and skin necrosis. Implant loss rates in the SBR 6.1% (*n* = 7) and PBR 9.4% (*n* = 5) were also not statistically significant (*p* = 0.99). The hemovac duration period was significantly longer in the SBR (14.93 ± 5.57 days) group than in the PBR group (11.09 ± 4.82 days) (*p* < 0.01). However, post-operative pain scores are similar between two groups, although it is not clear whether this was due to the effect of postoperative patient-controlled analgesia. *Conclusions:* A SBR is a commonly used procedure with various advantages, but there are many problems due to damage to the normal pectoralis major muscle. According to the results of our study, the PBR group had a shorter hemovac duration period compared to the SBR group, although there was no significant difference in complication rate. A PBR is a simple and safe technique allowing early discharge without increasing the incidence of long-term complications.

## 1. Introduction

Implant-based breast reconstruction (IBBR) is the most commonly used reconstruction modality after mastectomy. In 2019, in the United States, 88,005 IBBRs were performed, while 19,233 autologous tissue reconstructions were carried out [1]. The implant based reconstruction method can be divided into dual-plane subpectoral breast reconstruction (SBR) and prepectoral breast reconstruction (PBR) depending on where the breast implant is located. At the beginning of breast reconstruction, PBR without acellular dermal matrices (ADMs) was associated with an increase in the risk of complications such as rippling and long-term capsular contracture, and implant loss because of thin breast skin flap after mastectomy. These led to a submuscular approach where the implants were completely covered by the pectoralis major muscles to compensate for the shortcomings of PBR.

Insertion of the implant under the pectoralis and serratus muscles has been considered the safest technique, but it has the following limitations: limited projection and control of the lower pole to imitate the natural curvature of the breast and additional morbidity according to serratus elevation. With the advent of ADMs, dual plane SBR has become the most frequently performed surgical breast reconstruction procedure for the past 20 years. ADMs are biotechnologically engineered collagen layers that helps create an internal supportive and flexible structure to hold the implant in place [2,3]. They have served many purposes in SBR. Many surgeons developed techniques for using ADMs to support a cohesive gel implant with low complication rates like as seroma, infection, implant loss [4,5], and good aesthetic outcomes [6,7]. However, the risk of animation deformity [8], with contraction of the pectoralis muscle and postoperative pain [9], remains a potential consequence that can significantly affect quality of life [10].

Recently, an issue of a reverse shift from subpectoral to prepectoral placement has arisen because of the increase of the thickness and quality of skin flaps due to improved mastectomy flap dissection and development of ADM [11]. The benefits of PBR include not only elimination of animation deformity and muscle spasm, but optimal positioning of the device on the chest wall in accordance with the medial border of the mastectomy. However, statistics compiled by the American Society of Plastic Surgeons have demonstrated that 90% of the prosthetic reconstruction performed in the United States is in the way of 2 stages [1]. For this reason, little is known regarding the influence of implant placement in a direct-to-implant (DTI) reconstructive approach. The aim of this study was to describe outcomes and complications between PBR and SBR in patients who had undergone mastectomy followed by immediate DTI breast reconstruction.

## 2. Materials and Methods

### 2.1. Study Design

We performed a retrospective review of medical records from patients who underwent immediate unilateral DTI reconstruction with an acellular dermal matrix after a total mastectomy for breast cancer at our institution between February 2015 and February 2020. Patients who had a previous breast procedure, including breast conserving surgery for previous malignancy and a history of radiation therapy, were excluded from this study. This study was was approved by the institutional review board of Ewha Womans University Mokdong Hospital (No. 2020-07-047; approved date: 7 September 2020). In this study, patient demographics, history of postoperative radiation treatment, and pre–postoperative chemotherapy were compared between the dual-plane and prepectoral cohorts to assess uniformity of the patient populations in the two groups (prepectoral vs. subpectoral). In addition, intraoperative and postoperative factors such as resected specimen weight, implant size, and ADM size were recorded.

Outcome variables assessed for the two groups included development of complications, need for re-operation, total number of days before Jackson-Pratt (JP) drain removal, and pain scores. Presence of the following complications was evaluated: hematoma, seroma, cellulitis, infections, flap necrosis, and need for re-operation. Seroma was considered present when the collected fluid was confirmed by aspiration of serous fluid. Infection was based on clinical symptoms such as redness, swelling, and local hotness that required intravenous antibiotics and/or surgical intervention. Flap necrosis included any cases ranging from healing by secondary intention to revision surgery.

### 2.2. Surgical Technique

Mastectomy was performed by the general surgery (GS) team including skin-sparing mastectomy (SSM) and nipple-sparing mastectomy (NSM). NSM was considered first, unless there was preoperative evidence of malignant involvement of the nipples. The GS team excised the nipple if the intraoperative frozen biopsy of tissue under the nipple showed malignant cells during the operation. In case the nipple could not be preserved, it was defined as SSM. Lateral straight radial incisions and peri-areolar incisions were used, depending on location and size of the cancer. In all cases, great care was taken to maintain a well-vascularized subcutaneous tissue layer to ensure adequate blood supply to promote wound healing and ADM integration through neovascularization.

Single-stage reconstruction was performed for all patients who underwent mastectomy. After completion of the mastectomy, all procedures were performed by a single reconstructive plastic surgeon (SE.H) using a standardized operative technique. Patients were placed into either the dual-plane submuscular group or the prepectoral group, based on the surgeon’s intraoperative assessment and a thorough discussion with patients considering their desire and financial burden. In the subpectoral group, the inferior costal origin of the pectoralis major muscle was detached to create a dual plane. For cases of prepectoral implant placement, no muscle manipulation was performed.

In all cases, we used MegaDerm (L&C Bio, Seoul, Republic of Korea) or CG CryoDerm (CGBio Co., Seoul, Republic of Korea) as an ADM material for breast reconstruction. In the subpectoral group, after positioning the customized ADM between the pectoralis major muscle and inframammary fold, fixation was performed with 2-0 Vicryl suture (Ethicon Inc., Somerville, NJ, USA). The wounds were irrigated with gentamicin, cetrazole, and the chosen ADM was used to provide coverage for the inferior pole of the implant. Anatomical implants (Mentor Worldwide LLC, Santa Barbara, CA, USA) are prepared on the back table and soaked in triple antibiotic solution, and then were placed in the subpectoral–ADM pocket. In the prepectoral group, the cavity is irrigated and gloves changed before inserting the implant. The permanent implant is wrapped to cover the entire anterior surface and as much of the posterior surface as possible with a single piece of ADM (16 × 16 cm or 18 × 18 cm product) or two sheets of ADM (8 × 16 cm products) using a Vicryl suture. An appropriately sized cohesive silicone gel implant wrapped by ADM is placed in the prepectoral position.

Two drains were placed and skin flaps were closed over the pectoralis muscle in a standard fashion. Drains were removed when the amount of drainage was less than 30 mL/day for 2 consecutive days.

### 2.3. Pain Assessment

Patients were educated about visual analogue scale (VAS) scores on the day before the surgery. Postoperative pain-related data were collected at 12 h, 24 h, and 7 days from the time the patient left the operating room after surgery [10]. Every patient received short-acting intravenous narcotics during surgery and postoperative patient controlled analgesia for 48 h. Pain scores were recorded whereby patients rate their pain quantitatively on a Likert scale from 0 (no pain) to 10 (excruciating pain), which was given to the patient at the time of patient education. A VAS score of 5 was defined as the degree of pain at which the patient found it difficult to sleep or rest without additional pain control, and a VAS score of 10 was defined as pain as severe as death.

### 2.4. Statistical Analysis 

For statistical analysis between the prepectoral and subpectoral group, t-tests were used to analyze continuous variable differences. Categorical data were presented as a percentage and analyzed using the chi-square test, and for samples smaller than 5 data points, the Fisher exact test was used. In addition, hierarchical regression analysis was performed after controlling demographic characteristics such as age, BMI, resection specimen weight, and ADM size to determine whether prepectoral or subpectoral insertion of the implant affects the hemovac duration and postoperative pain score. All statistical analyses were performed using IBM SPSS Version 25 (IBM Corp., Armonk, NY, USA). Data were expressed as mean ± standard deviation (SD) for continuous variables.

## 3. Results

### 3.1. Patient Demographics and Operative Data

Over the duration of this study period, one-hundred sixty seven DTI reconstructions were performed: 68.3 percent were subpectoral (*n* = 114) and 31.7 percent were prepectoral plane (*n* = 53). Mean age and presence of comorbidities were not significantly different between the two cohorts. Mean body mass index was significantly larger for the prepectoral patient group than subpectoral patient group. The two groups were similar with regard to oncologic characteristics, including indication for mastectomy, mastectomy operation performed, chemotherapy before or after mastectomy, and adjuvant radiation therapy (Table 1).

Implant profile use was primarily anatomical (Style 311; Mentor, Inc., Santa Barbara, California) for both subpectoral and prepectoral. Average resection size was comparable between cohorts (subpectoral, 302.2 ± 173.0 g (IQR 24.0–349.3 g); prepectoral, 285.9 ± 116.9 g (IQR 62.0–354.0 g); *p* = 0.57) and implant volume was similar (subpectoral, 268.1 ± 103.0 cc (IQR 90.0–335.0 cc); prepectoral, 249.0 ± 104.8cc (IQR 90.0–350.0 cc); *p* = 0.27). A significantly larger size of ADM was used in SBR than PBR (subpectoral, 119.7 ± 36.6 cm^2^ (IQR 48.0–128.0 cm^2^); prepectoral, 261.7 ± 71.1 cm^2^ (IQR 96.0–324.0 cm^2^); *p* < 0.01).

### 3.2. Surgical Outcome and Complications 

The average days until drain removed were shorter in the prepectoral group than in the subpectoral group, and this was statistically significant (subpectoral, 14.93 ± 5.57 days (IQR 6–18 days); prepectoral, 11.09 ± 4.82 days (IQR 6–13 days); *p* < 0.01). Mean pain scores for the first 12 hours (subpectoral, 4.10 ± 1.28 (IQR 1–4); prepectoral, 4.49 ± 1.93 (IQR 1–6); *p* = 0.18), 24 hours after surgery (subpectoral, 2.36 ± 1.38 (IQR 1–2); prepectoral, 2.66 ± 1.82 (IQR 0–4); *p* = 0.29), and 7 days postoperatively (subpectoral, 0.80 ± 1.07 (IQR 0–1); prepectoral, 1.08 ± 1.19 (IQR 0–2); *p* = 0.14) were similar for both cohorts.

There were no significant differences in total complication rate between the two groups and no differences in any of the individual complications measured (Table 2). This included no statistically significant differences in rates of seroma (subpectoral, *n* = 14 (12.3%); prepectoral, *n* = 6 (11.3%); *p* = 0.86), infection (subpectoral, *n* = 11 (9.6%); prepectoral, *n* = 5 (9.4%); *p* = 0.97), mastectomy flap necrosis (subpectoral, *n* = 10 (8.8%); prepectoral, *n* = 2 (3.8%); *p* = 0.34), capsular contracture (subpectoral, *n* = 4 (3.5%); prepectoral, *n* = 2 (3.8%); *p* > 0.99), and explantation (subpectoral, *n* = 7 (6.1%); prepectoral, *n* = 5 (9.4%); *p* > 0.99). Both cohorts were well matched in safety profile.

We examined the results of hierarchical regression analysis after controlling demographic characteristics to determine whether prepectoral or subpectoral insertion of implants affects hemovac duration and postoperative pain score (Table 3). During statistical analysis, ADM size was excluded due to multicollinearity. In Model 1, age, BMI, and resection weight were set as control variables to determine the effect on hemovac duration and pain score. In Model 2, an insertion plane was added to Model 1 as an independent variable to investigate whether the hemovac duration and pain score were affected when the insertion plane was changed, even after demographic characteristics were controlled.

In the analysis of variance (ANOVA) test, the regression model was appropriate for hemovac duration only in Model 1 with *ΔF* = 9.857 (*p* < 0.001) and Model 2 with *ΔF* = 15.202 (*p* < 0.001). On the other hand, in Models 1 and 2 with pain scores at 12 hours, 24 hours, and 7 days, the regression model was not suitable. In both Models 1 and 2, the tolerance was more than 0.1 and the VIF was less than 10, confirming that there was no multicollinearity problem between variables. As a result of the t-test for the regression coefficient of Model 2, it was found that insertion of the implant into the prepectoral or subpectoral after demographic characteristics control with t = 5.157 (*p* < 0.001) had a statistically significant effect on the hemovac duration.

There were eight cases of complete device loss (subpectoral, *n* = 4; prepectoral, *n* = 4), while four implants (subpectoral, *n* = 3; prepectoral, *n* = 1) required downsizing to placement of a smaller size implant. Of the eight cases of complete device loss, four patients (subpectoral, *n* = 1; prepectoral, *n* = 3) were reconstructed with a delayed deep inferior epigastric perforator free flap.

## 4. Discussion

In the 1960s, breast reconstruction was performed with implants in the subcutaneous space [12]. Complications were common, including implant rippling, step-offs, skin necrosis, implant exposure, and capsular contracture [13,14]. In 1980s, to mitigate common complications of prepectoral insertion without ADM, the reconstructive technique was modified to provide additional soft tissue coverage and the implant was moved to a submuscular position [15]. When the full muscle coverage technique was introduced, implants were covered by both the pectoralis major muscle and the partial muscle, not just the pectoralis major. However, the main muscle was pectoralis major, which was not fixed to the chest wall, so it often migrated superiorly.

These problems were addressed by the advent of ADMs. ADMs are human-, bovine-, or porcine-derived biotechnologically engineered tissues that have served a numerous purposes across surgical subspecialties [16]. Tissue processing removes cellular antigens that can generate an immunological response while maintaining a structural matrix that promotes angiogenesis and tissue regeneration. In 2006, Salzberg published his experience using ADM in immediate breast reconstruction, and ADM became an important ingredient in breast reconstruction [3]. Afterward, the dual plane approach of ADM and covering the implant with the pectoralis major muscle was commonly performed. While cosmetic outcomes of the ADM-assisted dual-plane subpectoral approach have generally been quite good with limited postoperative complications, this type of reconstruction is associated with injury-induced muscular impairment, and the potential for breast animation deformity that can significantly affect quality of life [17,18].

ADM provides an additional layer of tissue between implant and skin that may reduce many of the complications seen when implants were placed directly under the skin [19,20]. Over the past decade, the advancement of ADM products has enabled prepectoral insertion to decrease the risks of overstretching the lower pole of the breast, and capsular contracture [21,22]. As it functioned as a barrier between the mastectomy skin and the implant, ADM has made it possible to return to PBR. Kim et al. reported that the levels of myofibroblasts were significantly lower in ADM capsules than in submuscular capsules [23]. With advances in oncologic surgical technique such as the NSM with optimal skin flap, improved implants, and ADMs, prepectoral breast reconstruction has become feasible. It quickly achieved a high degree of aesthetic satisfaction. However, the literature still lacks a reproducible demonstration of the physical advantages of this approach compared to the SBR technique [24].

Since prepectoral breast reconstruction has been resurrected, there have been several techniques used to cover the implant with ADM. In vivo techniques are commonly performed following completion of the mastectomy to cover just the anterior aspect of the device. Alternatively, the ex vivo full wrapping method entails complete coverage of both the anterior as well as the posterior surfaces of the implant with ADM before inset [25]. In this study, the implant was completely wrapped with ADM ex vivo (Figure 1). A single piece of 16 × 16 cm or 18 × 18 cm piece of ADM was fenestrated and all the sutures are on the posterior aspect of the construct. This full wrapping method allows IBBRs with extremely short operative times providing reproducible outcomes and excellent cosmesis [26].

We compared outcomes and complications between SBR and PBR with ADM in DTI breast reconstruction (Figure 2 and Figure 3). Our complication rates are similar to other authors that have reported on ADM–assisted IBBR [6,22,27]. Nadeem et al. was reported that the complete wrap is not only more expensive owing to its use of a larger sheet of ADM, but data have shown higher incidence of implant loss, capsular contracture, seroma, and flap necrosis [28]. However, in our study, there was no statistically significant difference in rates of complications between the PBR of the full wrapping method and SBR. It demonstrates that prepectoral implant placement and complete coverage with acellular dermal matrix is a safe technique. When doing breast reconstruction, it is common to insert a hemovac drain to prevent seroma or hematoma and maintain it until the amount is reduced below a certain level. According to our study, the prepectoral single stage DTI breast reconstruction group had a shorter hemovac duration period compared to SBR without increasing the incidence of seroma or hematoma. The reason is not clear yet, but it may be due to less invasive techniques by sparing the pectoralis muscle. Because our institution has a protocol that the patient is discharged on the day after hemovac is removed, the shorter hemovac duration period leads to a decrease in the postoperative length of inpatient hospital stay. The aesthetic results after surgery were also excellent in both PBR and SBR.

Our study did not demonstrate any difference in postoperative pain between groups, which is contrary to perceptions that PBR leads to a reduction in postoperative pain. We acknowledge that it is widely assumed that prepectoral reconstruction is less painful. Our results suggest that this may not be the case, and we hypothesize that the majority of the pain originates from the mastectomy and axillary procedures and that raising the pectoralis major is only a small component of perceived pain. Or we can hypothesize that the pain score difference was masked by the effects of postoperative patient controlled analgesia (PCA) device applied for 48 h after surgery. Therefore, large-scale prospective studies without postoperative PCA may be required.

When surgery is performed using the PBR of the full wrapping method, it is inevitable to use a large area of ADM, which can be costly to the patient. However, DTI PBR with complete ADM coverage presents an opportunity to improve upon current reconstructive methods for select mastectomy patients. Because it eliminates the need for any manipulation of the chest wall muscles, prepectoral breast reconstruction carries many potential advantages for patients. Current and newer silicone implants inserted with biological matrices in the prepectoral plane will allow plastic surgeons to potentially innovate the aesthetics of breast reconstruction [26,29]. Under the condition that the skin flap is not too thin and the patient does not refuse for cost reasons, prepectoral plane is the anatomical space the plastic surgeons should favor if human anatomy allows us to obtain aesthetical and functional outcomes [18].

Our outcomes suggest that immediate prepectoral IBBR with complete ADM coverage is a feasible and safe technique. This study can be utilized when considering reconstructive options and counseling patients on potential differences between the two techniques.

This study has limitations, given the methodology and assessed outcomes. As a retrospective study, there is an element of selection bias in the patients that were selected for a surgery type that cannot be avoided. However, this study from a single institution has relatively larger sample size for both subpectoral and prepectoral implant placement for IBBR.

## 5. Conclusions

An SBR is a commonly used procedure with various advantages, but there are many problems due to damage to the normal pectoralis major muscle. According to the results of our study, the PBR group had a shorter hemovac duration period compared to SBR group, although there was no significant difference in complication rate. A PBR is a simple and safe technique allowing early discharge without increasing the incidence of long-term complications.

## Figures and Tables

**Figure 1 medicina-56-00537-f001:**
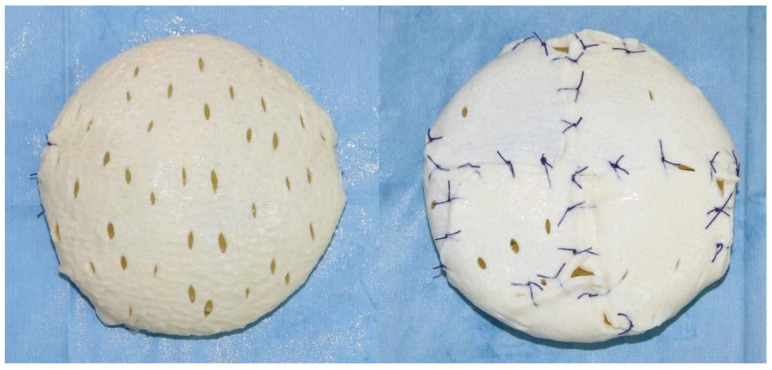
Ex vivo Full Wrapping Method.

**Figure 2 medicina-56-00537-f002:**
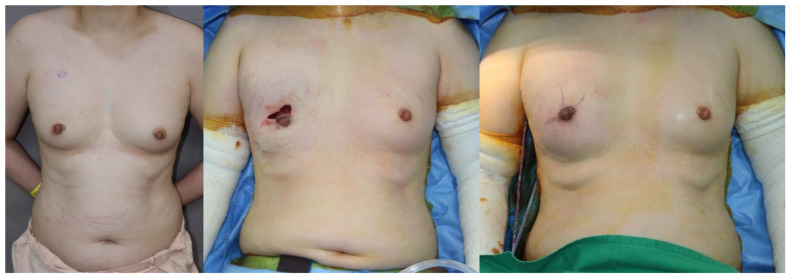
A case image of a patient with prepectoral insertion. A 41-year-old woman with a diagnosis of right breast cancer. (**left**) Preoperative image, (**center**) intraoperative image, (**right**) postoperative image.

**Figure 3 medicina-56-00537-f003:**
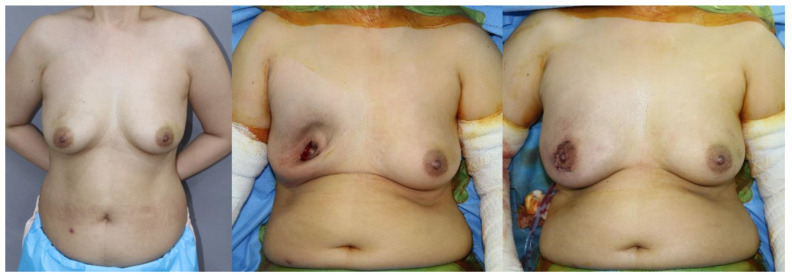
A case image of a patient with subpectoral insertion. A 44-year-old woman with a diagnosis of right breast cancer. (**left**) Preoperative image, (**center**) intraoperative image, (**right**) postoperative image.

**Table 1 medicina-56-00537-t001:** Clinical and surgical characteristics.

	Prepectoral	Subpectoral	*p*-Value
No. of patients (%)	53 (31.7)	114 (68.3)	
Age, mean ± SD, yrs	47.68 ± 7.45	46.56 ± 9.65	0.41
BMI, mean ± SD, kg/m^2^	23.92 ± 3.61	22.65 ± 2.81	0.01
Hypertension (%)	3 (5.7)	14 (12.3)	0.19
Diabetes (%)	2 (3.8)	3 (2.6)	0.65 *
Adjuvant chemotherapy (%)	17 (32.1)	49 (43.0)	0.18
Neoadjuvant chemotherapy (%)	3 (5.7)	12 (10.5)	0.39 *
Adjuvant radiation therapy (%)	6 (11.3)	21 (18.4)	0.25
Cancer laterality			0.87
No. of right (%)	30 (56.6)	63 (55.3)	
No. of left (%)	23 (43.3)	51 (44.7)	
Cancer stage			0.40
No. of stage I (%)	39 (73.6)	76 (66.7)	
No. of stage II (%)	11 (20.8)	24 (21.0)	
No. of stage III (%)	3 (5.7)	14 (10.2)	
No. of stage IV (%)	0 (0)	0	
Mastectomy type			0.99
No. of nipple-sparing (%)	47 (88.7)	101 (88.6)	
No. of skin-sparing (%)	6 (11.3)	13 (11.4)	
Mastectomy specimen weight, mean ± SD, g	285.9 ± 116.9	302.2 ± 173.0	0.57
Inserted implant volume, mean ± SD, cc	249.0 ± 104.8	268.1 ± 103.0	0.27
Inserted ADM size, mean ± SD, cm^2^	261.7 ± 71.1	119.7 ± 36.6	<0.01 †

SD, standard deviation; BMI, body mass index; * Between-group comparison was performed using Fisher’s exact test.; † Values are statistically significant.

**Table 2 medicina-56-00537-t002:** Surgical outcomes and complications.

	Prepectoral	Subpectoral	*p*-Value
Average days until drain removed, mean ± SD, days	11.09 ± 4.82	14.93 ± 5.57	<0.01 †
Pain scale, mean ± SD			
At 12 hrs after surgery	4.49 ± 1.93	4.10 ± 1.28	0.18
At 24 hrs after surgery	2.66 ± 1.82	2.36 ± 1.38	0.29
At 7 days after surgery	1.08 ± 1.19	0.80 ± 1.07	0.14
Complication			
Seroma, *n* (%)	6 (11.3)	14 (12.3)	0.86
Infection, *n* (%)	5 (9.4)	1 (9.6)	0.97
Hematoma, *n* (%)	0 (0)	1 (0.9)	>0.99 *
Skin necrosis, *n* (%)	2 (3.8)	4 (8.8)	0.34 *
Capsular contracture, *n* (%)	2 (3.8)	4 (3.5)	>0.99 *
Implant loss, *n* (%)	5 (9.4)	7 (6.1)	0.52 *

SD, standard deviation; * Between-group comparison was performed using Fisher’s exact test; † Values are statistically significant.

**Table 3 medicina-56-00537-t003:** Effect of insertion plane on hemovac duration and pain score after controlling demographic characteristics.

	Model 1				Model 2			
Hemovac Duration	*β*	t	*p*	VIF	*β*	t	*p*	VIF
Constant		1.068	0.287			−1.857	0.065	
Age	0.045	0.597	0.551	1.098	0.039	0.562	0.575	1.099
BMI	0.142	1.649	0.101	1.435	0.242	2.935	0.004	1.519
Resection weight	0.296	3.545	0.001	1.339	0.231	2.945	0.004	1.374
Insertion plane					0.356	5.157	<0.001 †	1.064
*ΔF*(*p*)	9.857 (<0.001 †)				15.202 (<0.001 †)			
*R*^2^ (adj. *R*^2^)	0.154 (0.138)				0.273 (0.255)			
**Pain score at 12 h after surgery**								
Constant		3.729	<0.001			3.899	<0.001	
Age	0.023	0.278	0.781	1.098	0.025	0.301	0.764	1.099
BMI	0.053	0.565	0.573	1.435	0.021	0.221	0.825	1.519
Resection weight	−0.089	−0.980	0.328	1.339	−0.068	−0.748	0.456	1.374
Insertion plane					−0.113	−1.404	0.162	1.064
*ΔF*(*p*)	0.398 (0.755)				0.793 (0.531)			
*R*^2^ (adj. *R*^2^)	0.007 (−0.011)				0.019 (−0.005)			
**Pain score at 24 h after surgery**								
Constant		2.001	0.047			2.205	−0.029	
Age	−0.013	−0.157	0.876	1.098	−0.012	−0.141	0.888	1.099
BMI	0.076	0.811	0.418	1.435	0.054	0.561	0.576	1.519
Resection weight	−0.109	−1.206	0.229	1.339	−0.095	−1.037	0.301	1.374
Insertion plane					−0.078	−0.969	0.334	1.064
*ΔF*(*p*)	0.507 (0.678)				0.615 (0.653)			
*R*^2^ (adj. *R*^2^)	0.009 (−0.009)				0.015 (−0.009)			
**Pain score at 7 days after surgery**								
Constant		0.999	0.319			1.681	0.095	
Age	0.022	0.269	0.788	1.098	0.024	0.295	0.769	1.099
BMI	−0.003	−0.028	0.978	1.435	−0.037	−0.389	0.698	1.519
Resection weight	0.026	0.292	0.771	1.339	0.049	0.534	0.594	1.374
Insertion plane					−0.124	−1.539	0.126	1.064
*ΔF*(*p*)	0.059 (0.981)				0.637 (0.637)			
*R*^2^ (adj. *R*^2^)	0.001 (−0.017)				0.015 (−0.009)			

VIF, Variance Inflation Factor; † Values are statistically significant.

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
