# Peer review of "A Comparative Analysis between Subpectoral versus Prepectoral Single Stage Direct-to-Implant Breast Reconstruction"

_medicina, 2020, doi:10.3390/medicina56100537_

Round 1

Reviewer 1 Report

This work seeks to compare prepectoral breast vs subpectoral breast reconstruction. The authors review 5 years worth of their institution's data and compare results including patient demographics, comorbidities, chemotherapy, laterality of cancer, type of mastectomy, and impact volume. Further, they assess post-surgical pain scores, and complications. Overall, the work provides insight into the differences (lack there of) of the preprectoral vs subpectoral reconstruction. I have a few comments that have listed below:

1. Spell out ADMs in first use in Abstract and in line 42 on page 1

2. Page 2, Line 48-50 is an incomplete thought…please complete the sentence starting with With the advent of ADMs. Maybe dual plan SBR has become the most…

Methods:

3. These were all breast cancer patients, please clarify and if so, please add cancer stage to the analysis. 

4. Did you identify NSM to SSM conversion/add the number of NSM and SSM conversions to results?

5. How often were pain scores assessed?

6. How were comorbidities identified/define?

7. Your sample would support more stringent for some of your outcomes. For instance, you could use a regression analysis to look at the difference between the 2 groups with respect to the average days until the drain was removed, or pain scores while adjusting for age, BMI, and ADM volume. 

Results

7. Lines 156-157 on page 4 are essentially just a repeat of likes 152-155 on the same page. Would exclude.

Discussion

8. Page 5 line 179, a repeat of “the partial” at end of the sentence.

Author Response

Dear reviewer

Thank you for taking the time to review our manuscript.

  1. The first word we used in Abstract and main text, ADMs, was spelled out and the abbreviation was written in parentheses.
  2. We corrected that sentence to a complete sentence.
  3. It was clarified that the patients who underwent reconstructive surgery using implants were patients who underwent total mastectomy due to unilateral breast cancer. In addition, we add cancer stage to the analysis.
  4. In GS operation, whether to preserve or resect the nipple was determined according to the results of intraoperative biopsy. The GS team excised nipple, if intraoperative frozen biopsy of tissue under the nipple showed malignant cells during the operation. In case nipple could not be preserved, it was defined as SSM.
  5. Pain scores were recorded three times, one each at 12 hours, 24 hours, and 7 days from the time the patient left the operating room after surgery.
  6. Comorbidities such as hypertension and diabetes were removed as they were included in the patient demographics.
  7. Based on your opinion, we conducted research through hierarchical regression while controlling demographic characteristics such as age, BMI, and ADM size, resection weight and added Table 3 which summarizes the results.
  8. We deleted the repetitive content below.

‘Pain scores were recorded whereby patients rate their pain quantitatively using VAS scores at 12 hours, 24 hours, and 7 days from the time the patient left the operating room after surgery.’

Sincerely,

Seung Eun Hong, M.D., Ph.D., J.D.

Department of Plastic and Reconstructive Surgery, Mokdong Hospital, College of Medicine, Ewha Womans University

Anyangcheon-ro 1071, Yangcheon-gu

Seoul, Republic of Korea

[email protected]

Reviewer 2 Report

Nice paper. It would have been better to perform this kind of study as a prospective one. I think that the aesthetic outcome should also be evaluated, basically that is the key point of the different pockets. At least a follower up on that topic should be provided

Additionally the authors write " since the PBR needs to be completely wrapped..." Please cite any paper/study that describes wrapping as necessary. In my hands it is not necessary. Healing is faster without wrapping, aesthetic outcome is the same. Using more foreign material i.e. ADM has to be sufficiently substantiated. 

Author Response

Dear reviewer

Many thanks for your review of our article and your insightful comments.

  1. There were no cases in which the two different pocket groups raised significant issues about aesthetic problems after surgery, and also the surgeon did not feel a significant difference in aesthetic outcomes between different pockets. However, as you pointed out, it seems to be of great significance to objectively evaluate the aesthetic outcome. To address the problems pointed out in this study, we will do well-designed, prospective, randomized controlled trials including objective aesthetic outcome evaluation as soon as possible.
  2. Prepectoral implant-based breast reconstruction involving full implant coverage with ADM provided patients with high satisfaction for aesthetic outcomes[1] and surgeons with operative efficiency.[2] This contents has been added to the main text. In addition, due to the limited work space in nipple-sparing mastectomy pockets when small peri-areolar incision was used, we judged that reconstruction can be best done through ex vivo full wrapping with ADM. However, we agree with your opinion that PBR need not to be completely wrapped with ADM. Therefore, we revised that content to ‘when surgery was performed using the PBR of the full wrapping method,’.

  1. Downs, Ronald K., and Kellee Hedges. An alternative technique for immediate direct-to-implant breast reconstruction—a case series. Plastic and Reconstructive Surgery Global Open, 2016, 4.7.
  2. Sigalove, Steven. Options in Acellular Dermal Matrix–Device Assembly. Plastic and Reconstructive Surgery, 2017, 140.6S: 39S-42S.

Sincerely,

Seung Eun Hong, M.D., Ph.D., J.D.

Department of Plastic and Reconstructive Surgery, Mokdong Hospital, College of Medicine, Ewha Womans University

Anyangcheon-ro 1071, Yangcheon-gu

Seoul, Republic of Korea

[email protected]

Reviewer 3 Report

I really enjoyed reading this paper which was written very simply but made it clear.

In future, the effects of post-op radiotherapy and long-term complications.

Author Response

Dear reviewer

Many thanks for your review of our article and your insightful comments.

Based on your opinion, we would like to conduct research about the effects of post-op radiotherapy and long-term complications in the near future.

Sincerely,

Seung Eun Hong, M.D., Ph.D., J.D.

Department of Plastic and Reconstructive Surgery, Mokdong Hospital, College of Medicine, Ewha Womans University

Anyangcheon-ro 1071, Yangcheon-gu

Seoul, Republic of Korea

[email protected]

Round 2

Reviewer 2 Report

ok now